# Enhanced Expression of Glycolytic Enzymes and Succinate Dehydrogenase Complex Flavoprotein Subunit A by Mesothelin Promotes Glycolysis and Mitochondrial Respiration in Myeloblasts of Acute Myeloid Leukemia

**DOI:** 10.3390/ijms25042140

**Published:** 2024-02-10

**Authors:** Yunseon Jang, Jeong Suk Koh, Jung-Hyun Park, Suyoung Choi, Pham Thi Thuy Duong, Bu Yeon Heo, Sang Woo Lee, Jung Yeon Kim, Myung-Won Lee, Seok-Hwan Kim, Ik-Chan Song

**Affiliations:** 1Translational Immunology Institute, School of Medicine, Chungnam National University, Daejeon 35015, Republic of Korea; harhie@naver.com (Y.J.); jhpark76@cnu.ac.kr (J.-H.P.); 2Department of Internal Medicine, Chungnam National University Hospital, Daejeon 35015, Republic of Korea; goldjs2323@naver.com (J.S.K.); iyoo23@naver.com (M.-W.L.); 3Brain Korea 21 FOUR Project for Medical Science, School of Medicine, Chungnam National University, Daejeon 35015, Republic of Korea; jdd02287@naver.com (S.C.); phamduong290194@gmail.com (P.T.T.D.); xeyk1603@naver.com (B.Y.H.); 4Department of Medical Science, School of Medicine, Chungnam National University, Daejeon 35015, Republic of Korea; lleesy98@naver.com; 5Research Institute for Medical Science, School of Medicine, Chungnam National University, Daejeon 35015, Republic of Korea; jyk0210@cnu.ac.kr; 6Department of Surgery, Chungnam National University Hospital, Daejeon 35015, Republic of Korea

**Keywords:** mesothelin, acute myeloid leukemia, glycolysis, oxygen consumption rate

## Abstract

Acute myeloid leukemia (AML) is an aggressive malignancy characterized by rapid growth and uncontrolled proliferation of undifferentiated myeloid cells. Metabolic reprogramming is commonly observed in the bone marrow of AML patients, as leukemia cells require increased ATP supply to support disease progression. In this study, we examined the potential role of mesothelin as a metabolic modulator in myeloid cells in AML. Mesothelin is a well-known marker of solid tumors that promotes cancer cell proliferation and survival. We initially analyzed alterations in mesothelin expression in the myeloblast subpopulations, defined as SSC-Alow/CD45dim, obtained from the bone marrow of AML patients using flow cytometry. Our results showed overexpression of mesothelin in 34.8% of AML patients. Subsequently, metabolic changes in leukemia cells were evaluated by comparing the oxygen consumption rates (OCR) of bone marrow samples derived from adult AML patients. Notably, a higher OCR was observed in the mesothelin-positive compared to the mesothelin-low and non-expressing groups. Treatment with recombinant human mesothelin protein enhanced OCR and increased the mRNA expression of glycolytic enzymes and mitochondrial complex II in KG1α AML cells. Notably, siRNA targeting mesothelin in KG1α cells led to the reduction of glycolysis-related gene expression but had no effect on the mitochondrial complex gene. The collective results demonstrate that mesothelin induces metabolic changes in leukemia cells, facilitating the acquisition of a rapid supply of ATP for proliferation in AML. Therefore, the targeting of mesothelin presents a potentially promising approach to mitigating the progression of AML through the inhibition of glycolysis and mitochondrial respiration in myeloid cells.

## 1. Introduction

Acute myeloid leukemia (AML) is characterized by the uncontrolled proliferation of myeloblasts in the bone marrow, leading to a lack of cellular differentiation [1,2]. The growth of myeloblasts in AML is highly dependent on the rapid replenishment of adenosine triphosphate (ATP), which fuels their metabolic needs and drives proliferation. Metabolic reprogramming in myeloblasts and leukemic stem cells (LSCs) has been previously reported [3,4]. Myeloblasts and LSCs exhibit similar but distinct metabolic traits. The disruption in the citric acid cycle is observed in both myeloblasts and LSCs. Myeloblasts exhibit increased glycolysis and ROS production through the AKT and mTOR signaling pathways [5], while LSCs rely on oxidative phosphorylation for ATP generation [3]. Myeloblasts, being rapidly growing, are more susceptible to anti-proliferative drugs, whereas LSCs, characterized by slow growth and unique metabolic features, present a challenge in this regard. Myeloblasts show heightened glycolysis and ROS production, with increased intermediates of mitochondrial respiration to support their proliferation. In contrast, LSCs predominantly rely on oxidative phosphorylation rather than glycolysis or lactate production to obtain ATP [3,6,7]. 

The dysfunction of the mitochondrial complex is associated with poor prognosis in AML and, therefore, considered a potential therapeutic target. Tigecycline, an inhibitor of OXPHOS, has been shown to promote apoptosis of AML cells, and clinical trials are underway to evaluate the efficacy of therapeutic agents targeting the mitochondrial complex I [8]. Studies to date have shown that the efficacy of enasidenib, a drug targeting mutations in the mitochondrial isocitrate dehydrogenase (IDH)2 gene, is limited to less than 13% of patients with IDH2 mutations, which only accounts for a subset of AML cases characterized by unrestricted cell proliferation [9,10]. Similarly, venetoclax, an inhibitor of B-cell lymphoma 2 (Bcl-2), has limited effectiveness as monotherapy. Moreover, both drugs are associated with side-effects, such as cytopenia [5,11,12,13]. At present, the specific signaling pathways and marker proteins associated with changes in OXPHOS observed in the bone marrow of AML patients remain largely unknown, highlighting a crucial need to develop mitochondrial-modulating drugs that comprehensively target the identified metabolic pathways and exert beneficial effects on a larger proportion of AML patients. Such a drug would not only improve the efficacy of treatment but also minimize adverse effects, ultimately leading to better patient outcomes.

Overexpression of mesothelin has been reported in the bone marrow of 36% of acute myeloid leukemia (AML) patients [14,15]. Under normal conditions, mesothelin expression is limited to mesothelial cells lining the pericardium and pleura [16,17]. Mesothelin is widely recognized as a marker of aggressiveness in lung and pancreatic cancers [15,17,18,19]. Earlier studies demonstrated an increase in pancreatic cancer cell metastasis in mesothelin-overexpressing mice. Conversely, the knockout of mesothelin restored normal growth and blood cell counts [20,21]. Recently, mesothelin-targeted agents, such as the secreted phosphoprotein 1 (SS1P) immunotoxin and anti-mesothelin chimeric antigen receptor (CAR)-T cells, have entered clinical trials for mesothelioma and ovarian cancer [22]. These therapies have high safety but limited efficacy as monotherapy.

Mesothelin overexpressed in pancreatic cancer activates Akt signaling [16,23], which is reported to stimulate mitochondrial respiration [24]. Additionally, mesothelin induces the production of interleukin-6 (IL-6), B-cell lymphoma-extra large (Bcl-xL), and cyclin E, promoting cancer cell proliferation [16]. The protein exists in both membrane-bound and soluble forms and contains a CA125 binding domain, participating in CA125-mediated signaling to stimulate cell proliferation and survival [17,25]. In acute myeloid leukemia (AML), activation of phosphatidylinositol-3-kinase (PI3K)/Akt and mammalian target of rapamycin (mTOR) signaling pathways leads to enhanced glycolysis and increased production of reactive oxygen species (ROS) in myeloblasts [25,26]. This metabolic alteration supplies ATP for rapid cell proliferation by increasing the production of citric acid cycle intermediates [14,27,28]. Despite its well-established role in oncogenic signaling and solid tumor growth, the precise signaling pathways of mesothelin in hematopoietic malignancies remain unclear. Furthermore, the impact of mesothelin on metabolic reprogramming is yet to be established. 

In this study, we analyzed mesothelin expression in the bone marrow of adult AML patients and explored the associated metabolic changes, particularly mitochondrial respiration. The correlation between mesothelin expression and metabolic alterations was investigated by categorizing patients based on metabolic status. Our collective findings suggest that mesothelin acts as a metabolic modulator, promoting ATP production for continuous proliferation and the survival of myeloblasts in the bone marrow of AML patients.

## 2. Results

### 2.1. Myeloblasts from Patients with Acute Myeloid Leukemia Exhibit Varying Levels of Mesothelin Expression

Mesothelin is overexpressed in various solid tumors, such as lung, pancreatic, and ovarian cancers, and is shown to contribute to cancer cell proliferation, migration, and survival [16,19,20,21,25]. Conversely, under physiological conditions, mesothelin is exclusively detected in normal restrictive mesothelial cells. In the bone marrow of acute myeloid leukemia, we observed a significant increase in the population of myeloblasts exhibiting moderate expression of CD45 (CD45^dim^) and low side scatter (SSC-A^low^) (Figure 1A). Myeloblasts typically differentiate into monocytes while lymphoid stem cells differentiate into lymphocytes. However, in AML patients, the bone marrow reveals an abundant myeloblasts subpopulation without undergoing a differentiation process [1,2]. The AML patient bone marrow specimens demonstrated a notable absence of differentiated monocytes. In contrast, it is known that normal bone marrow comprised populations of CD45^high^/SSC-A^low^ lymphocytes, CD45^low^/SSC-A^low^ red blood cells, as well as hematogones [29]. Following induced remission therapy (IT), the population of myeloblast cells vanished, and the bone marrow exhibited the recovery of normal cell populations. We further investigated mesothelin expression in myeloblasts of AML patients with the aid of flow cytometry. Our findings revealed that among AML patients, the proportion of mesothelin-positive myeloblasts exhibiting CD45^dim^/SSC-A^low^ characteristics was highest (22.2%) before remission induction therapy. Mesothelin positivity normalized after treatment, as shown in Figure 1A. However, in the population of CD45^high^/SSC-A^low^ lymphocytes within the bone marrow of the same patient, mesothelin-positive cell populations were not detectable at diagnosis or after treatment (Figure 1). Subsequently, patients were categorized based on the intensity of mesothelin expression. Remarkably, the population of mesothelin-positive CD45^dim^/SSC-A^low^ myeloblasts ranged from 3.73% to 22.2% in 7 out of 22 AML patients at diagnosis, while the remaining 15 patients had <1.57% mesothelin-positive myeloblasts (Figure 1B,E), as confirmed via flow cytometry. Following induced remission therapy, the majority of patients showed a significant reduction in mesothelin-positive cells. In contrast to the CD45^dim^/SSC-A^low^ myeloblast population, mesothelin-positive cells were not detectable in the CD45^high^/SSC-A^high^ monocyte and mature lymphocyte populations (Figure 1C) in both untreated and treated samples, with the exception of one patient who had a mesothelin-positive cell population of 3.74% (Figure 1C). But our analysis revealed significant differences in the Mean Fluorescence Intensity (MFI) of mesothelin between the studied groups (Figure 1D,F). Characteristics of AML patients are listed in Table 1. The collective findings suggest that mesothelin is induced in myeloblasts and differentially expressed among AML patients. Furthermore, the elimination of the mesothelin-positive myeloblast population after induced remission therapy implies a crucial role of mesothelin in the malignancy of AML cells within the bone marrow.

### 2.2. Mitochondrial Respiration Is Enhanced in Bone Marrow of AML Patients with Mesothelin Positivity

AML myeloblasts have a high demand for ATP to support their continuous proliferation and survival [4,26]. Metabolic alterations, including changes in mitochondrial bioenergetics and glycolysis, commonly occur in AML [3,4,6,7,14,30]. Previous studies have shown enhanced biogenesis of mitochondria and activity of mitochondria complexes II and V in the bone marrow of pediatric AML patients [14]. In this study, we investigated whether similar mitochondrial and glycolytic changes occur in adult bone marrow. To assess mitochondrial respiration and glycolysis, we measured the oxygen consumption rate (OCR) of bone marrow using an XFe96 analyzer (Agilent, Santa Clara, CA, USA). The key parameters of mitochondrial respiration and glycolysis were determined by measuring baseline OCR and sequentially injecting oligomycin (a complex I inhibitor), carbonyl cyanide 3-chlorophenylhydrazone (CCCP, an uncoupler), and rotenone (an ATPase inhibitor). Spare respiratory capacity (SRC) was calculated as the difference between basal and maximal respiration, reflecting the capacity of cells to respond to stress. Notably, the comparison of AML patients with a high percentage of mesothelin-positive cells to those with no mesothelin expression or <1% mesothelin-positive cells revealed significantly higher OCR in the mesothelin-high group (Figure 2A). Overall, the mesothelin-high group exhibited higher levels of basal respiration, ATP production, non-mitochondrial OCR, maximal respiration, and spare respiratory capacity (SRC) relative to the mesothelin-low group with increases of 2.48-, 2.99-, 2.04-, 4.08-, and 6.12-fold, respectively (Figure 2B–F). However, SRC (%), coupling efficiency, and proton leak were comparable between the two groups (Figure 2G–I). Additionally, to assess glycolytic activity in the AML bone marrow of each group, we simultaneously measured the extracellular acidification rate (ECAR) alongside OCR. An increase in ECAR in the mesothelin-high group was observed at baseline and after the injection of mitochondrial inhibitors (Figure 2J). Our results indicate that both mitochondrial respiration and glycolysis are enhanced in AML bone marrow of the mesothelin-high compared to the mesothelin-low groups, suggesting that increased mesothelin expression is associated with metabolic alterations in AML bone marrow.

### 2.3. Enhancement of Mitochondrial Respiration and ATP Production in AML Cells via Mesothelin Stimulation

Mesothelin is overexpressed in solid tumors and plays a role in activating Akt/PI3K and mTOR signaling pathways, thereby promoting cancer cell proliferation and survival. The protein interacts with CA125 [17,25,31] and has been linked to Akt-induced IL-6 production associated with enhanced mitochondrial respiration in pancreatic cancer [5,16]. Moreover, cyclin E1 stimulation by mesothelin contributes to cancer cell proliferation and survival, which rely on ATP supply from both mitochondria and glycolysis [5,26]. Considering the increased oxygen consumption rate (OCR) and extracellular acidification rate (ECAR) in the bone marrow of the mesothelin-high group, we postulated that mesothelin modulates metabolism in AML. To establish the influence of mesothelin on AML, the KG1α AML cell line was treated with 0.25 μg/mL or 0.5 μg/mL recombinant human (rh) mesothelin peptide (rhMesothelin) for 1 h. Subsequently, OCR and ECAR were measured to exclude the specific environmental effects of complex signaling networks that impact mitochondrial respiration and glycolysis in the bone marrow of AML patients (Figure 3A). Notably, the average basal OCR of cells treated with rhMesothelin increased by 30.0% at a concentration of 0.25 μg/mL and 25.3% at a concentration of 0.5 μg/mL (Figure 3B) compared to vehicle-treated control cells. ATP production showed an increase of 19.0% and 14.5% (Figure 3C) in the 0.25 μg/mL and 0.5 μg/mL rhMesothelin-treated groups while non-mitochondrial respiration remained comparable between the groups (Figure 3D). Regarding maximal respiration, OCR increased by 21.8% and 16.7% in the presence of 0.25 μg/mL and 0.5 μg/mL rhMesothelin (Figure 3E), respectively, while parameters such as SRC, SRC (%), and coupling efficiency were not significantly altered (Figure 3F–H). Moreover, proton leak was 48.6% and 48.0% higher in cells treated with 0.25 and 0.5 μg/mL rhMesothelin, respectively (Figure 3I). In terms of glycolysis, we observed a significant increase in ECAR in KG1α cells after 1 h of incubation with rhMesothelin (Figure 3J). In addition, the NOMO-1 AML cell line was treated with 1 μg/mL rhMesothelin for 24 h and basal respiration and ATP production were increased by 20.4% and 23.0%, respectively. 

Interestingly, maximal respiration of KG1α cells was significantly reduced following mesothelin knockdown using siRNA, along with a decrease in spare respiratory capacity (Appendix A). However, ECAR of mesothelin knockdown cells remained unchanged (Appendix A). The relatively smaller change between negative control and siMesothelin-treated groups may be attributable to the remaining mesothelin. Mesothelin mRNA expression remained at 41.3%, even after siRNA treatment. Overall, our findings provide evidence that mesothelin modulates mitochondrial respiration and glycolysis of AML cells. Although the treatment of KG1α cells with rhMesothelin resulted in enhanced basal respiration, ATP production, and maximal respiration, consistent with the metabolic profile observed in mesothelin-high AML bone marrow, as depicted in Figure 2, we observed no significant effects of rhMesothelin on spare respiratory capacity (SRC) and non-mitochondrial OCR. In fact, ATP concentration in the media after the treatment of rhMesothelin (0, 0.25, and 0.5 μg/mL) was significantly increased as confirmed via a luciferase assay (Appendix A). Furthermore, mesothelin-high AML bone marrow exhibited a proton leak comparable to that of the mesothelin-low group, whereas a significant induction of proton leak was observed in the rhMesothelin-treated cells. It should be noted that since we did not perform the sorting of mesothelin-enriched myeloblasts and measured OCR and ECAR in the total bone marrow, which comprises ~90% myeloblasts and 10% monocytes and mature lymphocytes, the observed discrepancy in OCR parameters between mesothelin-high bone marrow and rhMesothelin-treated AML cell groups may have arisen as a result of these factors. The collective results clearly indicate that mesothelin alters cancer metabolism to facilitate a rapid production of ATP. 

### 2.4. Induction of Glycolytic Enzymes and Mitochondrial Complex II Gene Expression by Mesothelin in AML Cells

Mesothelin stimulates the Akt pathway in solid tumors [5,23], which is known to increase mitochondrial respiration [24]. In our experiments, rhMesothelin treatment altered the metabolic state of AML cells (Figure 3). In tumor cells, glycolysis serves as a primary source of ATP [26,32,33]. rhMesothelin treatment led to increased ECAR in AML cells, indicating enhanced glycolytic function influenced by glucose transporter and glycolytic pathway enzymes resulting in lactate production. Accordingly, we examined the mRNA levels of glucose transporter 1 (GLUT1), hexokinase 2 (HK2), phosphofructokinase (PFK), and lactate dehydrogenase A (LDHA) via quantitative reverse transcriptase PCR (qRT-PCR) in KG1α cells treated with 0.25 or 0.5 μg/mL rhMesothelin for 24 h. After 24 h of rhMesothelin treatment (0.5 μg/mL), mRNA levels of Glut1 and HK2 were significantly increased by 1.83- and 1.92-fold, respectively (Figure 4A,B). However, mRNA levels of PFK were not significantly different between the groups (Figure 4C). Regarding lactate production, the mRNA level of LDHA was elevated 1.57-fold at a rhMesothelin concentration of 0.5 μg/mL (Figure 4D). These findings demonstrate that mesothelin promotes glycolysis by inducing the mRNA expression of glycolytic enzymes in AML cells.

We additionally observed a significant increase in baseline OCR, ATP production, and maximal respiration of AML cells in response to rhMesothelin treatment (Figure 3). Previous reports indicate that increased succinate dehydrogenase complex flavoprotein subunit A (SDHA) activity enhances mitochondrial metabolism in tumor cells, leading to changes in basal and maximal OCR [28]. Accordingly, qPCR analysis was conducted to ascertain whether mesothelin affects the expression levels of mitochondrial complex genes in AML cells. To assess potential alterations in oxidative phosphorylation (OXPHOS) complex genes, we analyzed the mRNA expression levels of NADH dehydrogenase subunit 1 (ND1) of complex I (Figure 4E), *Sdha* of complex II (Figure 4F), Ubiquinol-Cytochrome C Reductase Core Protein 2 (UQCRC2) of complex III (Figure 4G), Cytochrome c oxidase I (COX1) of complex IV (Figure 4H), and ATP Synthase F1 Subunit Alpha (ATP5A1) of complex V (Figure 4I) in KG1α cells treated with rhMesothelin at concentrations of 0.25 or 0.5 μg/mL. The *Sdha* mRNA level was increased 1.36-fold after the incubation of rhMesothelin (0.5 µg/mL) (Figure 4F), suggesting that mesothelin enhances mitochondrial respiration by inducing complex II-associated gene expression. Notably, the significant induction of *Glut1*, *Hk2*, *Ldha*, and *Sdha* mRNA by rhMesothelin was observed only at the higher concentration of 0.5 μg/mL, while there was a tendency for increased expression without reaching significance at the lower concentration. This discrepancy may be attributed to the duration of gene transcription stimulated by rhMesothelin. The collective findings indicate that mesothelin markedly upregulates the transcription of *Sdha*, a constituent of mitochondrial complex II, thereby enhancing mitochondrial respiration and the expression of genes encoding glycolytic pathway-associated enzymes in AML cells. Next, we performed the knockdown of mesothelin using siRNA in KG1α cells. The mesothelin mRNA level was decreased by 58.7% after 48 h of siRNA transfection. Moreover, Glut1 and Hk2 mRNA levels were reduced by 43.3% and 50.5% in Mesothelin KD cells, respectively (Figure 5A–J). Our data suggest that mesothelin selectively induces glycolytic enzymes and the mitochondrial complex II gene *Sdha*. In addition, mesothelin enhanced both mitochondrial respiration and glycolysis, leading to increased ATP production in AML cells (Figure 5K).

### 2.5. Mesothelin Increases Mitochondrial Reactive Oxygen Species and Antioxidant Gene Expression

In general, the augmentation of mitochondrial respiration is anticipated to heighten reactive oxygen species (ROS) generation, as the mitochondrial respiratory chain constitutes a principal source of intracellular ROS production [34]. Following the confirmation of a significant enhancement in mitochondrial respiration due to rhMesothelin treatment, as demonstrated by oxygen consumption rate (OCR) measurements (Figure 3), we proceeded to investigate the impact of rhMesothelin on ROS production in AML cells. This evaluation utilized the fluorescent dye H_2_DCFDA for detecting cytosolic ROS and MitoSOX for detecting mitochondrial superoxide. Our findings revealed a 9.8% increase in mitochondrial ROS levels in rhMesothelin-treated cells compared to the control, while cytosolic ROS levels remained comparable, as assessed through flow cytometry (Figure 6A–D).

Given the documented induction of antioxidant gene expression by oncogenes to counteract oxidative stress in cancer cells [35,36], we posited that mesothelin may influence antioxidant enzyme expression in AML cells. We measured the mRNA level of Nrf2 (nuclear factor, erythroid 2 like 2) in KG1a AML cells, a transcription factor known to induce antioxidant-related genes [37]. To assess whether mesothelin alters the expression of ROS scavengers, we used quantitative reverse transcription-polymerase chain reaction (qRT-PCR) to quantify the expression of transcripts encoding antioxidant enzymes including Nrf2, Nqo1, Gpx1, and Gr, along with transcript levels of mitochondrial antioxidant enzymes including Sod1 and Sod2. We found that both Sod1 and Sod2 mRNA levels increased 1.6- and 2.2-fold, respectively (Figure 6E,F), and Gpx1 mRNA levels increased 1.41-fold. However, GR, Nrf2, and Nqo1 mRNA levels were not significantly altered by rhMesothelin treatment at a concentration of 0.5 µg/mL (Figure 6G–J). Collectively, these results suggest that mesothelin mitigates oxidative damage resulting from increased mitochondrial respiration in AML cells by promoting the expression of antioxidant enzymes.

## 3. Discussion

Acute myeloid leukemia (AML) is a hematological malignancy characterized by the uncontrolled proliferation of immature myeloblasts in the bone marrow [4,26]. Genetic mutations in AML are heterogeneous, resulting in varying treatment responses and difficulty in identifying specific targets for complete remission [38]. Recent studies indicate that abnormal metabolism, attributed to proteins that modulate metabolic processes, contributes to the proliferation and progression of AML myeloblasts [3,4,6,7,14,26]. In the current study, we investigated the metabolic alterations in adult AML bone marrow and identified a key modulator protein involved in mitochondrial respiration and the glycolytic pathway.

AML cells have a higher demand for ATP to support their rapid proliferation and survival and possess a larger mitochondrial content compared to normal hematopoietic cells [3,6,27]. Mitochondrial dysfunction and defects in mitochondrial proteins are associated with poor prognosis and chemotherapy resistance in AML [30,39]. Leukemic cells also tend to upregulate fatty acid oxidation through mitochondrial uncoupling [40]. Pharmaceutical compounds targeting the mitochondria and metabolism of AML blasts have been developed that are currently undergoing clinical trials [12,13,41]. Tigecycline, an OXPHOS inhibitor, induces apoptosis in AML cells by inhibiting mitochondrial complex I and etomoxir, an inhibitor of mitochondrial fatty acid oxidation, which reduces ATP production. L-deprenyl suppresses mitochondrial respiration and glycolysis, leading to AML cell death [1]. FDA-approved drugs, such as enasidenib and venetoclax, are only beneficial in a limited proportion of patients with mitochondrial genetic mutations in IDH2 and Bcl-2, respectively, which promote leukemogenesis [9,10,11,12,13]. Despite significant advances in these metabolism and mitochondrial-targeting approaches, unclear signaling pathways of the drug targets, limited applicability, and side effects continue to pose challenges in AML treatment. Accordingly, we focused on mitochondrial and cellular metabolism-modulating proteins with a view to identifying the target pathways in AML cells.

Mesothelin is upregulated in the bone marrow of one-third of pediatric AML patients and known for its oncogenic properties in solid tumors, including pancreatic and ovarian cancer [15,19,20,22,42]. Enhanced pancreatic tumor metastasis due to Akt signaling and mitochondrial respiration has been reported in mice with overexpression of mesothelin [16,20,21]. Furthermore, mesothelin promotes cancer cell survival and proliferation through the activation of cyclin E and IL-6 signaling pathways [16]. Considering that Akt and mTOR signaling enhance glycolysis in AML myeloblasts, we investigated whether mesothelin expression is augmented in adult AML bone marrow in association with metabolic alterations. To achieve this, AML bone marrow samples were grouped into mesothelin-high and -low groups and metabolic parameters compared between groups. Notably, the mesothelin-high group showed enhanced glycolysis and mitochondrial respiration. Moreover, mesothelin significantly induced the mRNA expression of glycolytic pathway-related enzymes and succinate dehydrogenase complex flavoprotein subunit A (SDHA) of mitochondrial complex II. These gene expression alterations potentially account for the increased ATP requirement necessary for rapid growth of AML cells.

It is important to acknowledge that our study did not solely sort myeloblasts for the measurement of OCR and ECAR. Since ~90% of cells in AML bone marrow comprise myeloblasts, the presence of other cell types (such as monocytes and mature lymphocytes) cannot be completely excluded. Therefore, sorting and performing measurements on myeloblast cells exclusively would yield more accurate and conclusive data, which may reveal potential differences between the results obtained from mesothelin-treated cells and the overall cell population. Further investigations to elucidate the role of mesothelin using additional approaches are warranted, such as utilizing mesothelin knockout mice. The analysis of the correlations among metabolic changes, mesothelin expression, protein, and mRNA levels in actual patient samples should provide clearer insights into the impact of mesothelin signaling on disease progression and prognosis. 

As previously reported, myeloblasts exhibit increased glycolysis and mitochondrial respiration induced by the AKT and mTOR signaling pathways. This heightened metabolic activity leads to the production of reactive oxygen species (ROS), contributing to the rapid growth and proliferation of myeloblasts. In contrast, LSCs rely on OXPHOS to generate ATP, rather than glycolysis and lactate production [3]. This metabolic distinction renders LSCs susceptible to ROS generated by the mitochondrial electron transporter, leading to cell death. A comparison of ROS after the treatment of rhMesothein or siMesothelin will provide valuable insights into mesothelin-related ROS production, shedding light on the potential role of mesothelin in modulating cellular responses to oxidative stress. 

Furthermore, it would be beneficial to assess the current status and limitations of mesothelin inhibitors, such as mesothelin-targeted CAR-T and its impact on metabolism regulation. In combination with metabolism-regulating drugs, the modulation of mesothelin may provide an effective treatment strategy for AML patients of all ages, thereby overcoming the low genetic mutation rates associated with conventional drugs. We additionally identified specific chromosomal relocations, such as histone–lysine N-methyltransferase 2 (KMT2) rearrangement (Table 1), in a number of patients with high mesothelin levels. The confirmation of metabolic changes in a larger sample size of patients possessing both genetic mutations and high mesothelin levels may serve as a marker for targeted metabolic therapies and prognostic prediction. In summary, we identified mesothelin as a key protein in AML cell metabolism that represents a promising target for AML therapy.

## 4. Materials and Methods

### 4.1. Cell Culture and Transfection

KG1α human acute myeloid leukemia (AML) cell was cultured in Iscove’s Modified Dulbecco’s Medium (IMDM, Welgene, Gyeongsan-si, Republic of Korea) containing 10% FBS (Hyclone, Waltham, MA, USA), 1% penicillin, and streptomycin (Hyclone, Waltham, MA, USA) at 37 °C under 5% CO_2_ and 21% O_2_ conditions. KG1α cells were transfected with siRNA oligo duplexes targeting human Mesothelin and negative control siRNA (ORIGENE, Rockville, MD, USA) using Lipofectamine 2000 (Invitrogen, Carlsbad, CA, USA).

### 4.2. Human Bone Marrow Samples

Bone marrow samples from 23 AML patients and 5 healthy individuals were obtained from the hematology/oncology department of Chungnam National University Hospital, Republic of Korea. The demographic characteristics of these patients are presented in Table 1. With reference to previous studies, experiments were performed on selected patients with AML with expected mesothelin expression, particularly those with KMT2A gene rearrangements and core binding factor fusions [14]. All samples were isolated with Lymphprep^TM^ (STEMCELL Tech. Vancouver, BC, Canada) solution and stored in LN2 tank until before analysis. Written informed consent was acquired from each outpatient involved in the study, and this study was conducted in accordance with the provisions of the Declaration of Helsinki. 

### 4.3. Flow Cytometry of Bone Marrow Samples

Frozen bone marrow stock was thawed and resuspended in RPMI1640 medium. The suspension was centrifuged at 300× *g* for 3 min and the supernatant discarded. Next, cells were treated with 100 μg/mL DNase I (Stemcell, Vancouver, BC, Canada) for 10 min, filtered through a 70 μm cell strainer (SPL, Gyeonggi, Republic of Korea), and incubated at 37 °C under 5% CO_2_ and 21% O_2_ for 16 h. Bone marrow cells (2–5 × 10^5^ cells in a 5 mL FACS tube) were rinsed with PBS. After centrifugation at 1000 rpm for 3 min, PBS was discarded, and 1 μM of live/dead dye (Thermo Fisher Scientific, Waltham, MA, USA) was added and incubated for 15 min in the dark. Next, cells were washed with 1% BSA (Gibco, Billings, MT, USA) for blocking, which was removed after centrifugation. A rabbit anti-human mesothelin-APC antibody (Abcam, Cambridge, UK) and rabbit anti-human CD45-FITC antibody (Abcam) were used to analyze cell surface expression of mesothelin in myeloblasts. Cells were incubated with the above antibodies for 40 min at 4 °C. Following a wash with 1% BSA buffer and centrifugation at 1000 rpm for 3 min, the remaining cells were resuspended in 1% BSA and filtered using a 100 μm cell strainer. Fluorescence was detected using a FACS Novo flow cytometer (BD Biosciences, Franklin Lakes, NJ, USA) with excitation/emission wavelengths of 485/535 nm for CD45-FITC, 550/580 nm for mesothelin-APC, and 510/580 nm for live/dead™ dye (Thermo Fisher Scientific). Values are presented as mean fluorescence intensity and the percentage of positive cells in the cell population determined using FlowJo software (v10.9, BD Biosciences, Franklin Lakes, NJ, USA).

### 4.4. Measurement of Oxygen Consumption Rate (OCR) and Extracellular Acidification Rate (ECAR)

Bone marrow samples from AML patients and healthy individuals were washed with Seahorse XF RPMI1640 media (pH 7.4, Agilent, Santa Clara, CA, USA) and plated in poly-D-lysine (Gibco, Billings, MT, USA)-coated 96-well microplate (2 × 10^5^ cells/well, Agilent, Santa Clara, CA, USA). KG1α (1 × 10^5^ cells/well) cells were incubated with or without rhMesothelin (0.25 or 0.5 μg/mL, Abcam, Cambridge, UK) for 1 h. After centrifugation, the medium was removed, and the cells were washed with Seahorse XF DMEM medium (pH 7.4). Subsequently, the cells were plated in poly-D-lysine-coated 96-well microplates. The baseline OCR of the bone marrow or KG1α cells was measured using an XFe96 analyzer (Seahorse, MA, USA). Mitochondrial inhibitors such as oligomycin A (20 µg/mL), an ATPase inhibitor (Sigma-Aldrich, Saint Louis, MO, USA), carbonyl cyanide 3-chlorophenylhydrazone (CCCP, 50 µM), an uncoupler (Sigma-Aldrich, Saint Louis, MO, USA), and rotenone (20 µM), a mitochondrial complex I inhibitor (Sigma-Aldrich, Saint Louis, MO, USA), were sequentially injected into each well, and OCR was measured after injection at 37 °C. ECAR was simultaneously measured with OCR using the XFe96 analyzer. OCR parameters were calculated using WAVE software (Agilent, Santa Clara, CA, USA).

### 4.5. RNA Isolation and Quantitative Real-Time PCR

KG1α (2 × 10^5^ cells per well of 6-well plate) cells incubated with or without rhMesothelin (0, 0.25, or 0.5 ug/mL) for 24 h were harvested, and total RNA was isolated using Trizol reagent (Invitrogen, Carlsbad, MA, USA). cDNA was synthesized using reverse transcription kit (Invitrogen, Carlsbad, MA, USA) according to the manufacturer’s instruction. cDNA (100 ng), forward and reverse primers (10 pmole), 2× SYBR Master Mix (Appliedbiosystems, Foster City, CA, USA), and distilled water were mixed, and the mRNA expression levels were determined using a CFX96 Real-time PCR detection system (Biorad, Irvine, CA, USA). The expression level of each gene was normalized to GAPDH and presented as relative expression. Primers used in this study: GLUT1, 5′-GCCCTGGATGTCCTATCTGA-3′ (forward) and 5′-CCCACGATGAAGTTTGAGGT-3′ (reverse); HK2, 5′-TAGGGCTTGAGAGCACCTGT-3′ (forward) and 5′-CCACACCCACTGTCACTTTG-3′ (reverse); PFK, 5′-GAAGAGCCCTTCGACATCAG-3′ (forward) and 5′-TCTTCCTGCAGTCAAACACG-3′ (reverse); LDHA, 5′-TGTGCCTGTATGGAGTGGAA-3′ (forward) and 5′-AGCACTCTCAACCACCTGCT-3′ (reverse); ND1, 5′-ATATGACGCACTCTCCCCTG-3′ (forward) and 5′-TGAGTTGGTCGTAGCGGAAT-3′ (reverse); SDHA, 5′-TAGTTGGGGCTACAGGTGTG-3′ (forward) and 5′-GGATCACTTGAGACCAGCCT-3′ (reverse); UQCRC2, 5′-TCTTGTCCATGCTGCTTTTG-3′ (forward) and 5′-CGAGGACATGCTGAAGAACA-3′ (reverse); COX1, 5′-ACGTTGTAGCCCACTTCCAC-3′ (forward) and 5′-TGGCGTAGGTTTGGTCTAGG-3′ (reverse); GAPDH, 5′-TGCCTCCTGCACCACCAACT-3′ (forward) and 5′-ACACGTTGGCAGTGGGGACA-3′ (reverse); Mesothelin, 5′-TACAAGAAGTGGGAGCTGGA-3′ (forward) and 5′-TTGTGGGTAGAGCTCATCCA-3′ (reverse).

### 4.6. Statistical Analysis

Statistical analysis of the data was conducted using Prism version 8 software (GraphPad, San Diego, CA, USA). The data are presented as mean ± SD (error bars). The significance of differences between the control and experimental groups was analyzed using a one-tailed Student’s *t*-test. A *p*-value < 0.05 was considered statistically significant.

## Figures and Tables

**Figure 1 ijms-25-02140-f001:**
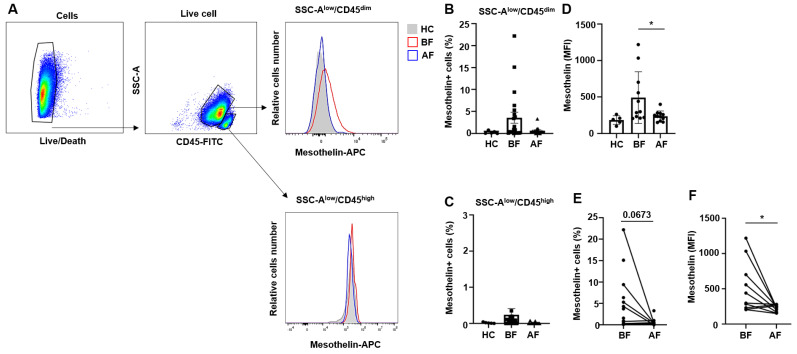
Expression of mesothelin in myeloblast subpopulation of AML patients at diagnosis and after induced therapy. (**A**) Gating strategy for AML bone marrow (2 × 10^5^ cells) based on CD45 expression. Myeloblast subsets are identified as CD45^dim^/SSC-A^low^. Mature lymphocyte subsets are identified as CD45^high^/SSC-A^low^. Histogram showing mesothelin surface expression in AML bone marrow at diagnosis (red), normal (gray area), and after treatment (blue) in the CD45^dim^/SSC-A^low^ myeloblast subsets (Upper). Histogram showing mesothelin expression in AML and normal CD45^high^/SSC-A^low^ subsets (Lower). (**B**) Percentage of mesothelin-positive cells in the CD45^dim^/SSC-A^low^ myeloblast subsets calculated using Flowjo software (v10.9), represented as a bar graph. (**C**) Percentage of mesothelin-positive cells in the CD45^high^/SSC-A^low^ population. (**D**) Mesothelin median fluorescence intensity (MFI) in CD45^dim^/SSC-A^low^ myeloblast subsets. Values are presented as mean ± SD (bars) (* *p* < 0.05, vs. corresponding controls). (**E**) Paired *t*-test of BF and AF in mesothelin-positive cells percentage. (**F**) Paired *t*-test of BF and AF in mesothelin MFI (* *p* < 0.05). HC, Healthy control; BF, Before induced-therapy (IT); AF, after IT.

**Figure 2 ijms-25-02140-f002:**
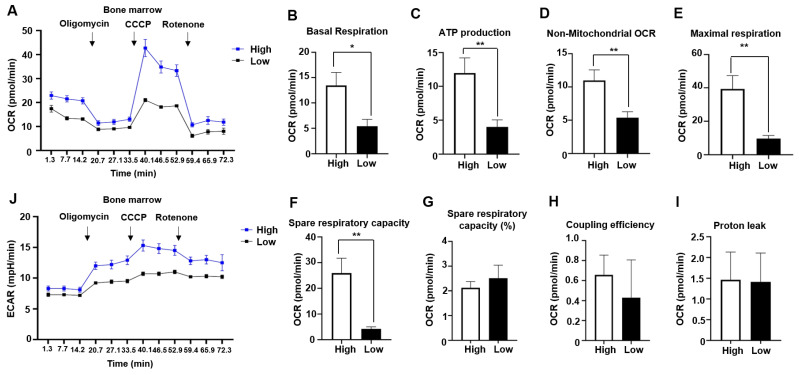
Enhanced oxygen consumption rate and extracellular acidification rate in the bone marrow of patients with mesothelin-high acute myeloid leukemia. (**A**) OCR of AML bone marrow (1 × 10^5^ cells/well) obtained at diagnosis, grouped by mesothelin expression levels, was measured after sequential injection of mitochondrial inhibitors, including oligomycin, CCCP, or rotenone. (**B**–**I**) Baseline OCR, ATP production, non-mitochondrial OCR, maximal respiration, spare respiratory capacity (SRC), SRC (%), coupling efficiency, and proton leak were calculated using WAVE software(v2.6). (**J**) ECAR was measured using the XFe96 analyzer simultaneously with OCR measurements. Values are presented as mean ± SD (bars) (* *p* < 0.05, ** *p* < 0.01 vs. corresponding controls). The black line indicates OCR or ECAR of the mesothelin-low group, while the blue line represents the mesothelin-high group.

**Figure 3 ijms-25-02140-f003:**
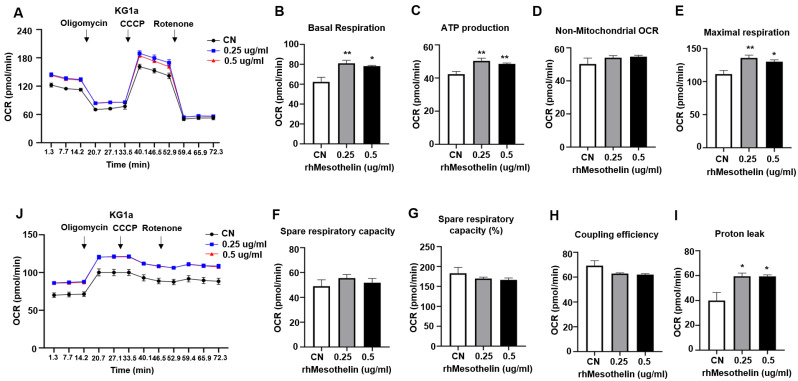
Mesothelin treatment increased oxygen consumption rate and glycolysis in AML cells. (**A**) KG1α (2 × 10^5^ cells/well) cells incubated with different concentrations (0, 0.25, or 0.5 μg/mL) of rhMesothelin for 1 h; OCR was measured after sequential injection of mitochondrial inhibitors. (**B**–**I**) Values of OCR parameters were calculated using WAVE software, and the values are presented as mean ± SD (bars) (* *p* < 0.05, ** *p* < 0.01 vs. corresponding controls). (**J**) ECAR was measured with XFe96 analyzer along with OCR after mitochondrial inhibitor injection. The black line indicates OCR or ECAR of control KG1α cells, the blue line represents the 0.25 μg/mL rhMesothelin-treated group, and the red line represents the 0.5 μg/mL rhMesothelin-treated group.

**Figure 4 ijms-25-02140-f004:**
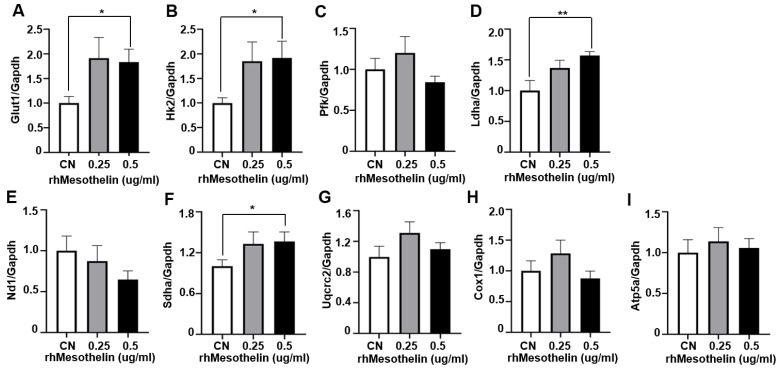
Effects of recombinant human mesothelin treatment on glycolysis and mitochondrial complex gene in KG-1a acute myeloid leukemia cells. (**A**–**I**) KG1α cells were incubated in media containing rhMesothelin (0, 0.25, or 0.5 μg/mL) for 24 h. mRNA expression of glycolysis-related enzyme gene (**A**–**D**) and mitochondrial OXPHOS complex gene (**E**–**I**) was assessed using qPCR analysis. Values were normalized to GAPDH levels and presented as mean ± SD (bars) (* *p* < 0.05, ** *p* < 0.01 vs. corresponding controls).

**Figure 5 ijms-25-02140-f005:**
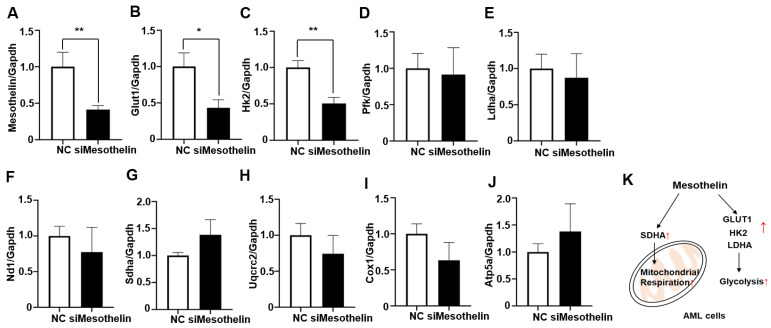
Effects of mesothelin knockdown on the expression of glycolysis and mitochondrial complex gene mRNA in KG-1a cells and schematic representation illustrating the modulation of AML cell metabolism by mesothelin. A-J, KG1α cells were incubated in media containing siRNA targeting mesothelin for 72 h. mRNA expression level of human mesothelin (**A**), glycolysis-related gene (**B**–**E**), and mitochondrial OXPHOS complex gene (**F**–**J**) was determined using qPCR analysis. Values are normalized to GAPDH levels and presented as mean ± SD (bars) (* *p* < 0.05, ** *p* < 0.01 vs. corresponding controls). (**K**) Illustration of metabolic modulation by mesothelin in human AML. Red arrow represents upregulation.

**Figure 6 ijms-25-02140-f006:**
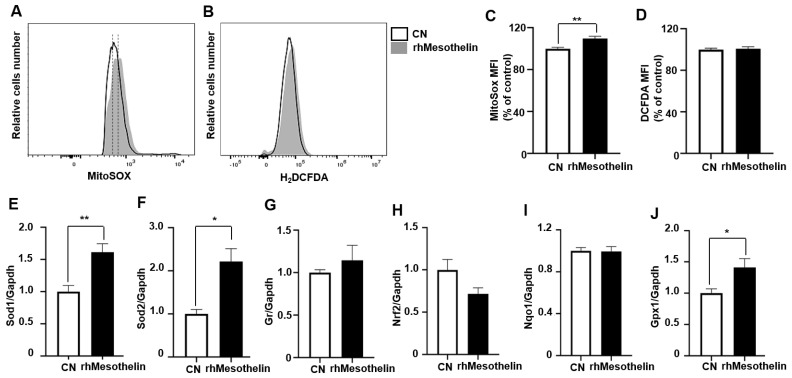
Mesothelin induces mitochondrial antioxidant enzyme expression. (**A**−**D**), KG1α cells were incubated in media containing rhMesothelin (0.5 μg/mL). Cells were stained with MitoSOX^TM^ or H_2_DCFDA, and fluorescence intensity was measured via flowcytometry. Mitochondrial ROS was determined by measuring relative cell number of MitoSOX^TM^-stained cells and H_2_DCFD-stained cells (**A**,**B**). Dashed line represents median value of relative cells number. (**C**,**D**) Median fluorescence intensity (MFI) values are presented as a mean ± SD (bars) (** *p* < 0.01 vs. corresponding controls). (**E**–**J**), mRNA expression level of human antioxidant enzyme genes was determined using qPCR analysis. Values are normalized to GAPDH level and presented as mean ± SD (bars) (* *p* < 0.05, ** *p* < 0.01 vs. corresponding controls).

**Table 1 ijms-25-02140-t001:** Baseline characteristics of patients in AML (*N* = 22).

Age, Median (Range)	59 (23–78)
Gender, M:F (%)	13:09
ELN classification	
AML with recurrent genetic abnormalities	21 (95.5%)
AML with mutated TP53	0 (0.0%)
AML with MR gene mutations	0 (0.0%)
AML with MR cytogenetic abnormalities	0 (0.0%)
AML, NOS	1 (4.5%)
NCCN risk stratification	
Favorable	7 (31.8%)
Intermediate	11 (50.0%)
Poor	4 (18.2%)
Peripheral blood tests, median (range)	
White blood cell count, ×10^3^/uL	5.61 (0.0095–163.9)
Hemoglobin (g/dL)	8.9 (5.3–12.8)
Platelet count, ×10^3^/uL	50 (9.0–266)
Absolute neutrophil count, ×10^3^/uL	0.60 (0.1–7.6)
LDH (IU/L)	1033 (389–11,000)
BM blast, median (range), %	70.0 (22.0–95.2)
Cytogenetics	
inv (16)/t (16;16)	6 (27.3%)
t (8;21)	9 (40.9%)
11q23/KMT2A rearrangements	4 (18.2%)
Somatic mutations	
FLT3-ITD	4 (18.2%)
FLT3-TKD	2 (9.1%)
NPM1	2 (9.1%)
DNMT3A	1 (4.5%)
IDH2	1 (4.5%)
TET2	2 (9.1%)
NRAS	5 (22.7%)
WT1	1 (4.5%)
PTPN11	1 (4.5%)
KIT	8 (36.4%)
U2AF1	1 (4.5%)
KRAS	1 (4.5%)
ASXL1	1 (4.5%)
SMC3	2 (9.1%)
SRSF2	1 (4.5%)
CBL	2 (9.1%)
KMT2C	2 (9.1%)
NOTCH2	2 (9.1%)
Disease status at end of induction (evaluable)	
1st CR	16 (72.7%)
2nd CR	0 (0.0%)
3rd CR	0 (0.0%)
Persistent	2 (9.1%)

ELN, European Leukemia Network; MR, myelodysplasia-related; LDH, lactate dehydrogenase; BM, bone marrow; NCCN, National Comprehensive Cancer Network; CR, complete remission.

## Data Availability

The datasets generated during and/or analyzed during the current study are available from the corresponding author on reasonable request.

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
