# Peer review of "Enhanced Expression of Glycolytic Enzymes and Succinate Dehydrogenase Complex Flavoprotein Subunit A by Mesothelin Promotes Glycolysis and Mitochondrial Respiration in Myeloblasts of Acute Myeloid Leukemia"

_ijms, 2024, doi:10.3390/ijms25042140_

Round 1
Reviewer 1 Report
Comments and Suggestions for Authors
The authors have established a novel observation of increased mesothelin expression with increased oxygen consumption rate (OCR). The authors can significantly increase the significance of the study by performing the following experiments:
Major:
1) Inclusion of one or more cells lines in addition to KG1alpha to replicate the key findings of the study will be valuable.
2) Use of knockout cell lines, instead of siRNA (since the inhibition is about 60% and transient). If knocking out mesothelin is fatal it would be good observation that could strengthen the observations of the study.
3) Since, according to the study, OCR leads to increased ATP generation which results in progression and survival of AML, it would be prudent to show ATP levels along with cell death and cell proliferation assay in mesothelin knockout cells.
4) Some of the observations like the decreases in mesothelin cells after induction therapy, if not statistically significant, should be clearly stated.
Minor:
1) Myeloblasts and hematogones have both been used interchangeably to define the SSC-A low and CD45 dim population and the author need to clarify this ambiguity throughout the paper.
2) Western blotting to show activation of downward signaling molecules in the siRNA and rh-mesothelin studies will bolster the results of the study.
Author Response
Jan. 14. 2024
Dear Editor of International Journal of Molecular Sciences,
We would like to thank the reviewers for the detailed comments on our manuscript (ijms-2772009). Please find enclosed our revised manuscript, entitled “Enhanced Expression of Glycolytic Enzymes and Succinate Dehydrogenase Complex Flavoprotein Subunit A by Mesothelin Promotes Glycolysis and Mitochondrial Respiration in Myeloblasts of Acute Myeloid Leukemia” which we would like to submit for publication in International Journal of Molecular Sciences.
We are grateful to the reviewers and editors for their suggestions. We believe that the changes made in response to these suggestions have strengthened our manuscript, including revising our findings and their interpretation. We have revised our manuscript to answer the point-by-point response to all comments raised by the reviewers. The changed manuscript is noted by blue colored text.
Sincerely yours,
Please address all correspondence to:
Ik-Chan Song, M.D., Ph.D.
Assistant Professor
Division of Hematology/Oncology
Department of Internal Medicine
Chungnam National University
(35015) 282 Munhwa-ro, Jung-gu, Daejeon, South Korea
Office: +82-42-280-8381
H.P: +82-10-8641-9741
FAX: +82-42-257-5753
E-mail: petrosong@cnu.ac.kr, petrosong@cnuh.co.kr
Seok-Hwan Kim, M.D., Ph.D.
Division of Liver Transplantation and Hepatobiliary Surgery
Department of Surgery
Chungnam National University Hospital
Chungnam National University School of Medicine
282 Munhwa-ro, Jung-gu Daejeon,35015, Korea
H.P: +82-10-8220-4377
FAX: +82-42-257-8024
E-mail: kjxh7@naver.com

Reviewer 2 Report
Comments and Suggestions for Authors
In the manuscript “Enhanced Expression of Glycolytic Enzymes and Succinate Dehydrogenase Complex Flavoprotein Subunit A by Mesothelin Promotes Glycolysis and Mitochondrial Respiration in Myeloblasts of Acute Myeloid Leukemia” report on the expression of mesothelin in AML primary samples and KG1 cell line. They report on the mitochondrial respiration, ATP production and induction of glycolytic enzymes depending on the mesothelin expression.
Broad comments: Paper is readable but lack significant data and should be improved.
Specific comments:
1. „One significant metabolic alteration observed in the bone marrow of AML patients is increased mitochondrial biogenesis and oxidative phosphorylation” – please be precise whether it is in AML blasts or LSC, since it is well known that their biology differs.
2. Figure 1 “normal (blue), and after treatment (gray area)”.........and in the figure you state that the gray is healthy and blue is after induced therapy? You should clarify this.
3. Figure 1 - histograms usually have counts or normalized ratio on y-axis
4. Figure 1 – please state the MFI (meal fluorescence intensity) values of each of the cases and also statistical analysis. Did you use isotypic control? Did the autofluorescence of the cells change depending on the treatment? Include live/dead gating in Figure1
5. Figure 1 – CD45/SSC should differentiate between lymphocytes and monocytes – I can only see one population besides blasts.
6. Why did the mesothelin expression change in mononuclear cells (ly+mono) after treatment? How do the MFIs of blasts and ly+mono correlate? What is your cut-off for mesothelin positivity – in Figure 1 MFI of ly-mono is altogether higher than MFI of blast cells?
7. The authors should show results in more than one cell line
8. The authors should show ROS results, at least in cell lines
9. Figure 2 and Figure 3 are mixed
10. Is mesothelin promising target in only eliminating myeloblasts or it has effect in LSC?
11. What about the bone marrow samples of 5 healthy individuals? Why were dose bone marrow samples aquired from healthy individuals?
Comments on the Quality of English LanguageEnglish is sufficient.
Author Response

(The authors gave the same response as above.)

Round 2
Reviewer 2 Report
Comments and Suggestions for Authors
The authors failed to meet all the requirements to improve the value of the manuscript - eg. authors did not provide reactive oxygen species (ROS) results, as suggested in previous review.
Comments on the Quality of English LanguageExtensive editing of English language required
Author Response
Feb. 1. 2024
Dear Editor of International Journal of Molecular Sciences,
We would like to thank the reviewers for the detailed comments on our manuscript (ijms-2772009). Please find enclosed our revised manuscript, entitled “Enhanced Expression of Glycolytic Enzymes and Succinate Dehydrogenase Complex Flavoprotein Subunit A by Mesothelin Promotes Glycolysis and Mitochondrial Respiration in Myeloblasts of Acute Myeloid Leukemia” which we would like to submit for publication in International Journal of Molecular Sciences.
We are grateful to the reviewers and editors for their suggestions. We believe that the changes made in response to these suggestions have strengthened our manuscript, including revising our findings and their interpretation. We have revised our manuscript to answer the point-by-point response to all comments raised by the reviewers. The changed manuscript is noted by blue colored text.
Sincerely yours,
Please address all correspondence to:
Ik-Chan Song, M.D., Ph.D.
Assistant Professor
Division of Hematology/Oncology
Department of Internal Medicine
Chungnam National University
(35015) 282 Munhwa-ro, Jung-gu, Daejeon, South Korea
Office: +82-42-280-8381
H.P: +82-10-8641-9741
FAX: +82-42-257-5753
E-mail: petrosong@cnu.ac.kr, petrosong@cnuh.co.kr
Seok-Hwan Kim, M.D., Ph.D.
Division of Liver Transplantation and Hepatobiliary Surgery
Department of Surgery
Chungnam National University Hospital
Chungnam National University School of Medicine
282 Munhwa-ro, Jung-gu Daejeon,35015, Korea
H.P: +82-10-8220-4377
FAX: +82-42-257-8024
E-mail: kjxh7@naver.com

Round 3
Reviewer 2 Report
Comments and Suggestions for Authors
I have no further suggestions
Comments on the Quality of English LanguageMinor editing of English language required